# Susceptibility Testing by Volatile Organic Compound Detection Direct from Positive Blood Cultures: A Proof-of-Principle Laboratory Study

**DOI:** 10.3390/antibiotics11060705

**Published:** 2022-05-24

**Authors:** Sacha Daniëlle Kuil, Soemeja Hidad, Caroline Schneeberger, Pragya Singh, Paul Rhodes, Menno Douwe de Jong, Caroline Elisabeth Visser

**Affiliations:** 1Department of Medical Microbiology, Amsterdam Infection & Immunity Institute, Amsterdam UMC, University of Amsterdam, 1105 AZ Amsterdam, The Netherlands; s.hidad@amsterdamumc.nl (S.H.); c.schneeberger@amsterdamumc.nl (C.S.); m.d.dejong@amsterdamumc.nl (M.D.d.J.); c.e.visser@amsterdamumc.nl (C.E.V.); 2Specific Diagnostics, San Jose, CA 95134, USA; psingh@specificdx.com (P.S.); prhodes@specificdx.com (P.R.)

**Keywords:** direct antimicrobial susceptibility testing, blood cultures, phenotypic antimicrobial susceptibility testing, volatile organic compounds, rapid diagnostics

## Abstract

Background: Bacteria produce volatile organic compounds (VOCs) during growth, which can be detected by colorimetric sensor arrays (CSAs). The SpecifAST^®^ system (Specific Diagnostics) employs this technique to enable antibiotic susceptibility testing (AST) directly from blood cultures without prior subculture of isolates. The aim of this study was to compare the SpecifAST^®^ AST results and analysis time to the VITEK^®^2 (bioMérieux) system. Methods: In a 12-month single site prospective study, remnants of clinical positive monomicrobial blood cultures were combined with a series of antibiotic concentrations. Volatile emission was monitored at 37 °C via CSAs. Minimal Inhibitory Concentrations (MICs) of seven antimicrobial agents for *Enterobacterales, Staphylococcus*, and *Enterococcus* spp. were compared to VITEK^®^2 AST results. MICs were interpreted according to EUCAST clinical breakpoints. Performance was assessed by calculating agreement and discrepancy rates. Results: In total, 96 positive blood cultures containing *Enterobacterales, Staphylococcus*, and *Enterococcus* spp. were tested (269 bug–drug combinations). The categorical agreement of the SpecifAST^®^ system compared to the VITEK^®^2 system was 100% and 91% for Gram-negatives and Gram-positives, respectively. Errors among Gram-positives were from coagulase-negative staphylococci. Overall results were available in 3.1 h (±0.9 h) after growth detection without the need for subculture steps. Conclusion: The AST results based on VOC detection are promising and warrant further evaluation in studies with a larger sample of bacterial species and antimicrobials.

## 1. Introduction

Rapid antimicrobial susceptibility test (AST) results of bacteria causing bloodstream infections are essential to guide targeted antimicrobial therapy, as ineffective empirical treatment increases the risk for morbidity and mortality in bacteremia [1,2]. Phenotypic AST results are preferable given the poor correlation between genotypes and phenotypes, the high number of resistance genes, and the lack of information on Minimal Inhibitory Concentrations (MIC) for clinical decision-making [3,4]. Phenotypic AST is dependent on bacterial growth rate and the lag phase of microbial growth is one of the time-limiting factors [3]. Current AST results using automated growth-based systems (e.g., VITEK^®^2, bioMérieux, Inc., Durham, NC, USA and BD PHOENIX^TM^, Franklin Lakes, NJ, USA) often take at least two days after bacterial growth detection, partly due to the need for a subsequent subculture step, to standardize the initial inoculum size for reliable results [5,6,7,8,9]. AST results during a single working shift are not possible when a subculture step is required, like in current AST methods. Subcultures might be unnecessary when measuring viability indirectly, e.g., by volatile organic compound (VOC) detection for AST, thus potentially enabling AST results within a single working shift.

VOCs are small metabolites (<300 Da) produced during bacterial growth. Several functions of VOCs have been described involving the interaction between other physically separated micro-organisms, such as modulation of virulence or growth, by diffusing in water or air [10]. The group of VOCs is heterogeneous, involving different chemical classes such as alcohols, ketones, and benzenoids [11]. Various biosynthetic pathways are involved in the production of VOCs contributing to this high diversity. Some VOCs can be seen as common metabolites, produced by many different bacteria or bacterial groups, while other VOCs are specific to certain genera or even species [10]. Due to the unique fingerprint of VOCs in bacterial species, the differentiation between species is possible and has been studied using different detection methods, such as an electric nose, ion mobility spectrometry, and gas chromatography/mass spectrometry (GC/MS) [12,13,14,15,16]. Nevertheless, these techniques come with some challenges, such as the inability to identify and quantify VOCs (electric nose), the large, lab-based instruments (GC-MS), and high costs (GC-MS and ion mobility spectrometry) [17,18]. In contrast, colorimetric sensor array (CSA) is a technique that measures VOCs by a sensor with chemo responsive indicators, changing color when exposed to different VOCs. This results in a fast, inexpensive, portable, and simple to operate method [18,19]. Successful differentiation between bacterial species using CSA has been described [19,20]. Additionally, identification of different yeast species was described previously. with promising results [21]. Susceptibility testing based on VOC detection is not widely studied, but discrimination between susceptible and resistant *Staphylococcus aureus* and *Escherichia coli* strains (for oxacillin and ampicillin or gentamicin, respectively) have been described previously [22,23,24]. The SpecifAST^®^ system (Specific Diagnostics, CA, USA), a prototype of the RevealAST^TM^ system, measures VOCs of bacteria combined with a selection of antibacterial agents by CSA.

The aim of our study was to compare SpecifAST^®^ AST results and the corresponding analysis time to the VITEK^®^2 AST system directly from clinical positive blood cultures with *Enterobacterales*, *Staphylococcus*, and *Enterococcus* spp.

## 2. Methods

In a 12-month single site prospective laboratory study, the SpecifAST^®^ system was evaluated in comparison to the current standard practice, using remnant materials of positive blood cultures collected in routine clinical care.

### 2.1. Objectives

The primary objective of this study was to assess the performance of the SpecifAST^®^ system compared to the VITEK^®^2 system. Performance was measured by the categorical agreement, defined as agreement for the same categorical interpretation (susceptible (S), susceptible increased exposure (I), or resistant (R)) based on MIC values, in accordance with the 2021 European Committee on Antimicrobial Susceptibility Testing (EUCAST) breakpoints [25]. The analysis time in hours and number of Colony Forming Units per milliliter (CFU/mL) were measured as secondary objectives.

### 2.2. Sample Selection

From May 2017 until June 2018, blood cultures were selected for SpecifAST^®^ testing at the Medical Microbiology department of the Amsterdam University Medical Centers. Microbial growth in blood cultures was detected in the automated blood culture systems BACTEC™ FX (BACTEC; Becton Dickinson, Sparks, MD, USA) or BACT/ALERT^®^3 D (BacT/ALERT 3D; bioMérieux, Marcy, L’Étoile, France), followed by Gram-staining. Aerobic blood culture bottles containing Gram-negative rods or Gram-positive cocci were eligible for SpecifAST^®^ testing (Figure 1, SpecifAST®, Specific Diagnostics, San Jose, CA, USA). One positive blood culture was randomly selected by the study staff, once or twice a day, due to limited capacity of the SpecifAST^®^ system. Exclusion criteria were (1) positive blood cultures of the same patient enrolled in the past 30 days, to ensure enrolment of unique bacteremia episodes, or (2) polymicrobial blood cultures.

### 2.3. Sample Preparation for SpecifAST^®^ Testing

Blood cultures containing Gram-positive bacteria and Gram-negative bacteria were diluted 1:150 and 1:500, respectively, in cation adjusted Mueller Hinton II Broth (CA-MHB II). Samples were combined 1:1 with a range of antibiotics for SpecifAST^®^ testing (Table 1, Figure 2). *Enterobacterales* spp. were tested for susceptibility to cefotaxime, ciprofloxacin, and meropenem; *Enterococcus* spp. were tested for susceptibility to vancomycin and ampicillin; and *Staphylococcus* spp. were tested for susceptibility to cefoxitin, oxacillin, and ampicillin (Table 1). The SpecifAST^®^ software for VOC pattern analysis was trained for these three groups of (easily growing) bacterial species, which are also the most often identified in clinical blood cultures in our academic hospital. Each bug–drug combination was tested in triplicate, including a positive (no antibiotics) and negative control (no bacteria). Subcultures were performed to verify the number of CFU at the time of testing.

### 2.4. SpecifAST^®^ Testing

Colorimetric Sensor Array (CSA) caps (Specific Diagnostics) were used for detecting VOCs. Once the vials were inoculated with the bacterial suspension, a pre-assembled CSA cap was screwed on top of the vial. The cap contained six indicator spots which react to VOCs via color change in the presence of bacterial growth. The vials were inverted on a flat-bed scanner with the CSA caps facing the scanner. This assembly was incubated at 37 °C at 200 rpm, and VOC patterns were measured every 10 min. The total run time was 12 h, collecting a total of 72 images by the completion of the assay. An early (2018) version of Specific Diagnostics proprietary software was used to monitor the array of volatile-responsive sensors positioned in the headspace above each growing sample. Divergence of sensor responses in positive controls versus those with antibiotics were utilized to determine MICs.

### 2.5. Comparator Method

VITEK^®^2 (bioMérieux, Marcy, L’Étoile, France) AST was used as a comparator method (Figure 1). VITEK^®^2 AST cards N344, AST P567, and AST P586 were used for AST of *Enterobacterales*, *Staphylococcus*, and *Enterococcus* spp., respectively, based on matrix-assisted laser desorption-ionization/time-of-flight mass spectrometry (MALDI-TOF MS, Bruker Daltonics, BD, Bremen, Germany) results for bacterial identification after subculture on solid media.

### 2.6. Data Analysis

Positive blood cultures containing *Enterobacterales*, *Staphylococcus*, or *Enterococcus* spp. were eligible for analysis. Samples with errors leading to missing VITEK^®^2 or SpecifAST^®^ test results were excluded from the analysis. MIC values were interpreted and compared in accordance with the EUCAST clinical breakpoints [25]. For all *Staphylococcus* spp., cefoxitin 6 mg/L breakpoint was used. In bug–drug combinations without categorical agreement, discrepancy rates were calculated: very major error (VME; falsely susceptible), major error (ME; falsely resistant), and minor error (mE; susceptible, increased exposure versus susceptible or resistant) [26]. Numbers and percentages of agreement and corresponding error rates were calculated using IBM^®^ SPSS^®^ Statistics 26.

### 2.7. Ethical Considerations

This study was exempted from medical ethical review and informed consent as anonymized leftover materials were used. Specimens were only used when patients had not indicated any objection for anonymized research use of specimens according to local regulations.

## 3. Results

From May 2017 until June 2018, 136 aerobic positive clinical aerobic blood cultures containing Gram-positive cocci or Gram-negative rods were tested in the SpecifAST^®^ and VITEK^®^2 system. Twenty-five samples were excluded for analysis due to polymicrobial growth (6) or non-eligible bacterial species such as *Streptococcus* spp. or *Pseudomonas aeruginosa* (19) (Figure 3; Appendix A). Another 15 blood cultures were excluded since tests results (44 bug/drug combinations) were not available due to errors: 13 preparation errors, 26 SpecifAST^®^ technical errors, and 2 errors due to lack of bacterial growth in both SpecifAST^®^ and VITEK^®^2 testing (Appendix A).

A total of 96 positive blood cultures (269 bug–drug combinations) were included for analysis: 46 *Enterobacterales* spp. (47.9%), 40 *Staphylococcus* spp. (41.7%), and 10 *Enterococcus spp.* (10.4%). Eight different *Enterobacterales* spp., mostly *Escherichia coli* (*n* = 28, 60.9%), four different *Staphylococcus* spp. and two different *Enterococcus* spp., were identified (Table 2). No meropenem resistant *Enterobacterales* spp. or methicillin resistant *Staphylococcus aureus* were found (Table 3). One vancomycin resistant *Enterococcus* spp. was found. In four samples, species were identified as Extended Spectrum Beta Lactamase (ESBL) producing *Enterobacterales*.

### 3.1. Primary Outcome SpecifAST^®^ Performance

The overall categorical agreement between both AST methods for *Enterobacterales* spp. was 100% (134/134), as no discrepancies in interpretative results for ciprofloxacin, cefotaxime, and meropenem were found (Table 3). In *Staphylococcus* spp. the categorical agreement was lower: 90.5% (105/116), due to 5 very major errors (5/30, 16.7%) and 6 major errors (6/86, 7.0%). All five VME concerned *Staphylococcus epidermidis*, whereby four cefoxitin resistant strains (MIC 6 mg/L) and one oxacillin resistant strain (MIC > 2 mg/L) were reported as susceptible (MIC ≤ 0.25 mg/L) by SpecifAST^®^ (Appendix A). The six major errors concerned *Staphylococcus aureus* (2) and coagulase negative staphylococci, CoNS (4) with discrepancies in oxacillin and vancomycin susceptibilities (Appendix A). In *Enterococcus* spp. the overall categorical agreement was 94.7% due to one minor error (1/19). The overall sensitivity and specificity of the SpecifAST method compared to the VITEK2 was 91% (95%-CI: 83.1–98.9) and 97% (95%-CI: 94.8–99.2), Appendix A.

### 3.2. Secondary Outcomes: Analysis Time and Number of CFU

The mean analysis time was 3.1 h (+/−0.9 h) by the SpecifAST^®^ system compared to 10.5 h (2.6 +/−SD) by VITEK^®^2, which corresponds to a 70.5% reduction. A subculture step of 18–24 h was required prior VITEK^®^ testing, while this was not needed prior to SpecifAST^®^ testing. The time between bacterial growth detection in the automated blood culture systems and the initiation of the SpecifAST^®^ tests varied between 2.5 h and 38.5 h. Blood cultures were used directly for SpecifAST^®^ tests, without standardization of the inoculum, and the mean numbers of CFU per blood culture bottle were 10^8.80 ± 0.69^ and 10^8.75 ± 0.45^ for Gram-positives and Gram-negatives, respectively.

## 4. Discussion

The SpecifAST^®^ AST results in our proof-of-principle laboratory study are promising for *Enterobacterales* spp., as full categorical agreement was found. Given the small number of ciprofloxacin and cefotaxime resistant strains, and the lack of meropenem resistant strains, further studies are required to establish the performance of detecting resistance.

The categorical agreement for Gram-positives was acceptable (≥90%), but the error rates were high. All very major errors found in this study included oxacillin or cefoxitin resistance in *Staphylococcus epidermidis*. Beta-lactam resistance (such as oxacillin and cefoxitin resistance) in CoNS is very common [27,28] and mostly mediated by alteration of the penicillin-binding-protein (PBP2a), encoded by the MecA gene [29]. Detection of beta-lactam resistance in CoNS is complex, even in current AST methods, as resistance can be expressed solely in a small proportion of cells (heterotypic phenotype) [30,31,32]. An essential difference between the SpecifAST^®^ system and the VITEK^®^2 system is the use of a single or a few colonies from subcultures by the VITEK^®^2 system, while the SpecifAST^®^ system directly uses positive blood cultures. It could be hypothesized that VOC detection in the entire bacterial population represents the main expressed beta-lactam phenotype in CoNS. Therefore, SpecifAST^®^ results could hypothetically be more representative of the entire bacterial population present in the blood culture. For beta-lactam resistance detection in CoNS, further studies to assess whether AST of the entire bacterial population might be an advantage compared to AST of single colonies, which should include standard reference testing for beta-lactam resistance either by PBP2a culture colony test or MecA PCR [30].

Susceptibility testing based on VOCs as surrogates of viability is a new area of research. The correlation between VOCs and AST results in different combinations of bacteria and antibiotics may vary [3]. There are many factors that influence VOC levels, either directly, e.g., temperature or humidity (volatility), or indirectly, e.g., oxygen or the nutrient availability in a growth medium (bacterial metabolism) [10]. Further investigations should aim to standardize those factors to limited variability in results.

The time range between bacterial growth detection and SpecifAST^®^ testing was broad due to the laboratory operating hours. The corresponding inoculum sizes for SpecifAST^®^ testing also varied widely. It is notable that a high categorical agreement was found despite a wide range of CFUs in tested samples, as standardized inoculum sizes are required for most AST systems for reliable MIC values [33]. This might be due to the fact that SpecifAST^®^ detects changes in bacterial metabolic products instead of bacterial growth. By facilitating non-standardized initial inoculum sizes and omitting prior subculture steps, SpecifAST^®^ enables the rapid availability of test results, possibly facilitating same-day test results after bacterial growth detection. As we did not formally assess the correlation between inoculum size and AST results, the reliability of VOC-based AST without subculture steps should be assessed in future studies.

One of the limitations of this study was the comparison with the VITEK^®^2 system, instead of a reference method for AST, such as broth dilution [34,35]. The VITEK^®^2 system is our current clinical reference test like many clinical laboratories, which use less time-consuming and laborious phenotypic and genotypic AST tests [34,35]. Although the VITEK^®^2 system has proven to be a reliable AST method, it is possible that some of the discordant results were due to VITEK^®^2 errors. Further studies should therefore compare the SpecifAST^®^ system to a reference method.

Second, only a limited number and type of bacterial species and antibiotics were included in this study. Only *Enterobacterales*, *Staphylococcus*, and *Enterococcus* spp. were eligible for testing with the SpecifAST^®^ prototype. Future studies should assess AST results of other bacterial species and multiple antibiotics using the automated Reveal AST^TM^ system, which is an updated version of the SpecifAST^®^ system that enables parallel identification and susceptibility testing for a panel of antibacterial agents. Polymicrobial blood cultures were excluded from our analysis and some concerns exist related to the use of polymicrobial specimens. First, it is known that bacterial species can affect VOC production of other species, which might play a role in polymicrobial specimens [36]. Second, direct positive blood culture testing provides a combined resistance phenotype of all bacteria present in polymicrobial specimens, which might not be clinically relevant. AST results in polymicrobial specimens varied when using another rapid system that enables AST without prior subcultures, the Accelerate Pheno™ system (Accelerate Diagnostics, Tucson, AZ) [37], based on fluorescence in-situ hybridization [38,39]. Follow-up studies should assess whether AST results are reliable in various polymicrobial specimens.

Finally, this study was performed in a setting with low AMR levels. To evaluate the SpecifAST^®^ performance in detecting resistance, further studies in settings with higher AMR prevalence are required.

## 5. Conclusions

Measuring volatile organic compounds is a novelty for antimicrobial susceptibility testing. Our proof-of-principle results are promising, and further investigation is warranted to establish the reliability of results in multiple combinations of bacteria and antimicrobials.

## Figures and Tables

**Figure 1 antibiotics-11-00705-f001:**
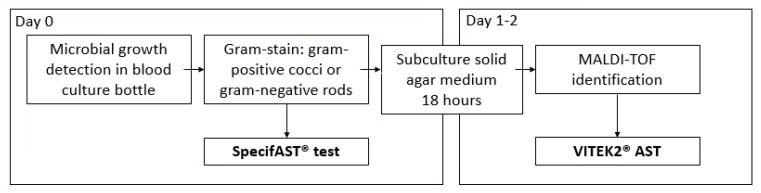
Study procedures MALDI-TOF: matrix-assisted laser desorption-ionization/time-of-flight mass spectrometry.

**Figure 2 antibiotics-11-00705-f002:**
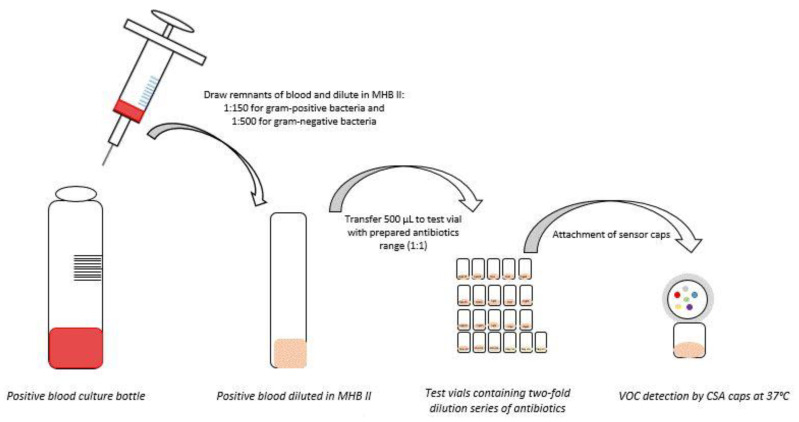
Sample preparation for SpecifAST^®^ testing MHB II: Muller Hinton II Broth; CSA: Colorimetric sensor array; VOC: Volatile organic compounds.

**Figure 3 antibiotics-11-00705-f003:**
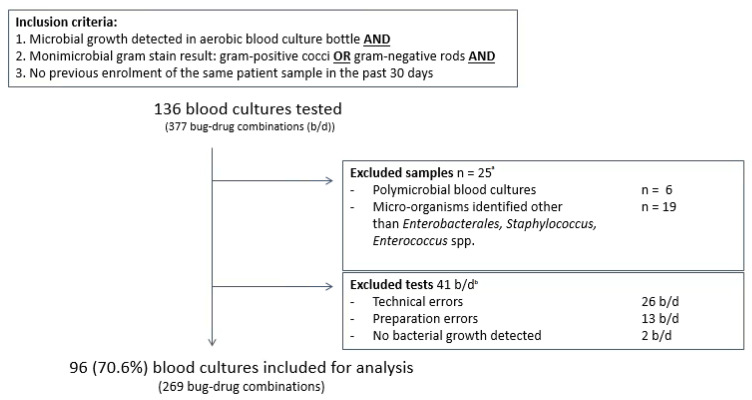
Baseline figure in and excluded samples. Inclusion criteria were microbial growth detection in aerobic blood culture bottles and monomicrobial gram stain result with gram-positive cocci or gram-negative rods and no previous enrolment of the same patient in the past 30 days. Exclusion criteria were polymicrobial blood cultures, micro-organisms, other than *Enterobacterales* spp., *Staphylococcus* spp. *or Enterococcus* spp., and errors leading to missing data. Error details are provided in Appendix A. B/d: bug–drug combinations. ^a^ 25 blood culture samples and all corresponding b/d combinations (67) were excluded from analysis based on polymicrobial growth or micro-organism identification result. ^b^ In 15 blood culture samples, all b/d combinations were excluded from analysis due to errors or absence of microbial growth. In 2 blood culture samples, one or more b/d combinations were excluded from analysis due to errors or absence of microbial growth. In total, 41 b/d combinations were excluded.

**Table 1 antibiotics-11-00705-t001:** Combinations of micro-organisms and antimicrobial agents (bug–drug combinations) and corresponding antibiotic concentration ranges tested.

Micro-Organism Group	Antimicrobial Agent	Specifast^®^ Concentration Range	VITEK2^®^ Concentration Range
*Enterobacterales* spp.	Ciprofloxacin Cefotaxime Meropenem	0.25–4 μg/mL 0.25–64 μg/mL 0.25–16 μg/mL	0.25–4 μg/mL 0.25–64 μg/mL 0.25–16 μg/mL
*Staphylococcus* spp.	Oxacillin + 2% NaCl ^a^ Vancomycin Cefoxitin	0.25–4 μg/mL 0.5–32 μg/mL 6 μg/mL	0.25–4 μg/mL 0.5–32 μg/mL 6 μg/mL
*Enterococcus* spp.	Ampicillin Vancomycin	2–32 μg/mL 0.5–64 μg/mL	2–32 μg/mL 0.5–32 μg/mL

NaCl: Sodium chloride; ^a^ The first seven *Staphylococcal* blood cultures were tested with solely oxacillin, in the consecutive samples Oxacillin with two percent of sodium chloride was used.

**Table 2 antibiotics-11-00705-t002:** Identified micro-organisms in included blood culture samples.

*Enterobacterales* spp.	*N* = 46
*Citrobacter freundii*	1 (2.2%)
*Klebsiella aerogenes*	1 (2.2%)
*Escherichia coli*	28 (60.9%)
*Klebsiella pneumoniae*	9 (19.6%)
*Klebsiella variicola*	3 (6.5%)
*Morganella morganii*	3 (6.5%)
*Proteus mirabilis*	1 (2.2%)
*Salmonella typhi*	1 (2.2%)
***Enterococcus* spp.**	***N* = 10**
*Enterococcus faecalis*	3 (30.0%)
*Enterococcus faecium*	7 (70%)
***Staphylococcus* spp.**	***N* = 40**
*Staphylococcus aureus*	12 (30.0%)
*Staphylococcus epidermidis*	13 (32.5%)
*Staphylococcus hominis*	11 (27.5%)
*Staphylococcus capitis*	4 (10.0%)

**Table 3 antibiotics-11-00705-t003:** Number and percentages of categorical and essential agreement between SpecifAST^®^ and VITEK2^®^ AST results, corresponding error rates and percentages, resistance rates, and time to test results per micro-organism group and antimicrobial agent.

	Antimicrobial Agent	Number of Clinical Blood Cultures Tested	Number of AB Tested (B/D)	CA	VME	ME	mE	R	TTR (h) SpecifAST Mean (SD)	TTR (h) VITEK2 ^a^ Mean (SD)
*Enterobacterales* spp.	Cefotaxime	45 (46.9%)	45 (16.7%)	45 (100%)	0 (0%)	0 (0%)	0 (0%)	7 (15.6%)		
	Ciprofloxacin	44 (45.8%)	44 (16.4%)	44 (100%)	0 (0%)	0 (0%)	0 (0%)	9 (20.5%)		
	Meropenem	45 (46.9%)	45 (16.7%)	45 (100%)	0 (0%)	0 (0%)	0 (0%)	0 (0%)		
	Total	46 (47.9%)	134 (49.8%)	134 (100%)	0 (0%)	0 (0%)	0 (0%)	16 (11.9%)	2.8 (0.6)	9.8 (2.6)
*Staphylococcus* spp.	Cefoxitin	38 (39.6%)	38 (14.1%)	34 (89.4%)	4 (25%)	0 (0%)	*n*.a.	16 (42.1%)		
	Oxacillin	39 (40.6%)	39 (14.5%)	34 (87.1%)	1 (7.1%)	4 (16%)	0 (0%)	14 (35.9%)		
	Vancomycin	39 (40.6%)	39 (14.5%)	37 (94.9%)	0 (0%)	2 (5.1%)	0 (0%)	0 (0%)		
	Total	40 (41.7%)	116 (43.1%)	105 (90.5%)	5 (16.7%)	6 (7.0%)	0 (0%)	30 (25.9%)	3.4 (1.1)	11.5 (2.4)
*Enterococcus* spp.	Ampicillin	10 (10.4%)	9 (3.3%)	8 (88.9%)	0 (0%)	0 (0%)	1 (11.1%)	5 (55.6%)		
	Vancomycin	10 (10.4%)	10 (3.7%)	10 (100%)	0 (0%)	0 (0%)	0 (0%)	1 (0.4%)		
	Total	10 (10.4%)	19 (7.1%)	18 (94.7%)	0 (0%)	0 (0%)	1 (1.7%)	6 (31.6%)	3.2 (0.5)	8.8 (1.6)
**Overall**		96 (100%)	269 (100%)	257 (95.5%)	5 (9.6%)	6 (2.8%)	1 (0.4%)	52 (19.3%)	3.1 (0.9)	10.5 (2.6)

Data are *n* (%) unless otherwise indicated. **^a^**12 b/d combinations missing VITEK2**^®^** TTR. AB: antibiotics; CA: categorical agreement; VME: very major error (the number of false susceptible b/d in SpecifAST**^®^,** divided by the number of resistant b/d in VITEK2**^®^**); ME: major error (the number of false resistant b/d in SpecifAST**^®^,** divided by the number of susceptible b/d in VITEK2**^®^**); mE: minor error (number of intermediate versus resistant or susceptible b/d, divided by the total number of b/d tested); NA: not available; R: resistance; TTR: time to test result.

## Data Availability

The dataset is fully accessible under CC0 license on figshare via DOI: 10.6084/m9.figshare.12052782.

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
