# Peer review of "Susceptibility Testing by Volatile Organic Compound Detection Direct from Positive Blood Cultures: A Proof-of-Principle Laboratory Study"

_antibiotics, 2022, doi:10.3390/antibiotics11060705_

Round 1

Reviewer 1 Report

The authors of this manuscript have provided a limited scoped, single location, lab-oriented proof-of-concept for the use of SpecifAST analysis method. Methods that can reduce the time lag, provide accurate results quicker, make it safer for the operator, and is also cost-effective are highly warranted. I did find the concept and the idea interesting and the method might find a broader application if its limitations are eliminated. This method can be used directly on the blood samples without the need to further culture the bacteria which makes it a faster and safer alternative. Use of this method was limited only to three bacterial species (for this study) which makes it slightly less intriguing. As the authors themselves record in the discussion section, this is a proof-of-concept study that needs to be further evaluated to make SpecifAST usable in the general analysis lab.

The most important advantage of this method is the quick and reliable detection and high level of agreement for the Gram-negative bacteria. The use of the colorimetric sensor array method may be considered another advantage of this method, provided the cost and time actually factors in its real-life use.

There are some major gaps in the study at this stage,

  1. High Error rates for categorical agreement for Gram-positive bacteria
  2. A considerable number of technical errors.
  3. Study was performed at a single location, which is less than ideal.

It would make the paper more instructive if the authors can provide a satisfactory answer to the following questions:

  1. In this study the application is limited to only three bacterial species, can this method be useful for other species or this is an intrinsic limitation of the method?
  2. The reason for selecting these particular bacterial species remained unknown.
  3. Detection of bacterial susceptibility in a polymicrobial culture possible by this method?

The paper as such may be accepted for publication in Antibiotics.

Author Response

The authors of this manuscript have provided a limited scoped, single location, lab-oriented proof-of-concept for the use of SpecifAST analysis method. Methods that can reduce the time lag, provide accurate results quicker, make it safer for the operator, and is also cost-effective are highly warranted. I did find the concept and the idea interesting and the method might find a broader application if its limitations are eliminated. This method can be used directly on the blood samples without the need to further culture the bacteria which makes it a faster and safer alternative. Use of this method was limited only to three bacterial species (for this study) which makes it slightly less intriguing. As the authors themselves record in the discussion section, this is a proof-of-concept study that needs to be further evaluated to make SpecifAST usable in the general analysis lab.

The most important advantage of this method is the quick and reliable detection and high level of agreement for the Gram-negative bacteria. The use of the colorimetric sensor array method may be considered another advantage of this method, provided the cost and time actually factors in its real-life use.

There are some major gaps in the study at this stage,

  • High Error rates for categorical agreement for Gram-positive bacteria
  • A considerable number of technical errors.
  • Study was performed at a single location, which is less than ideal.

It would make the paper more instructive if the authors can provide a satisfactory answer to the following questions:

Reviewer #1 Question #1 and #2: In this study the application is limited to only three bacterial species, can this method be useful for other species or this is an intrinsic limitation of the method. The reason for selecting these particular bacterial species remained unknown.

Reviewer #1 Answer #1 and #2: For the proof-of-concept phase we selected three easy growing groups of micro-organisms which were successfully evaluated in a pre-clinical phase to train the software for volatile organic compound pattern analysis. In our academical hospital those micro-organisms are identified most often in clinical blood cultures. We fully agree that this method should be evaluated in various bacterial species identified in clinical blood cultures (such as streptococci and Pseudomonas aeruginosa). Meanwhile, the RevealAST is developed for a broader range of micro-organisms (and antibiotics). Therefore evaluation of other bacterial species will follow. We have included the rationale for selecting those bacterial strains in the materials and methods section, lines 148 – 151:

The SpecifAST software for VOC pattern analysis was trained for these three groups of (easily growing) bacterial species, which are also the most often identified in clinical blood cultures in our academic hospital.”

Reviewer #1 Question #3: Detection of bacterial susceptibility in a polymicrobial culture possible by this method?

Reviewer #1 Answer #3: SpecifAST method cannot determine susceptibility in polymicrobial cultures. Most rapid direct from positive blood culture systems have some risk that the sample is polymicrobial, but often the second strain is a contaminant or present at a far lower concentration, for instance because it is slower growing than the dominant strain. For this reason, all rapid systems, including SpecifAST, require a purity plate to ascertain monomicrobial versus polymicrobial status. In the discussion section (row 282-290) we have briefly explained that polymicrobial cultures were excluded from this study. We believe that the risk associated with those polymicrobial samples is small, in that they represent a small fraction of the positive blood cultures that go undetected as polymicrobial after Gram stain, and that the rapid AST results provided by this method is of very substantial utility as it defines drug/concentration pairs sufficient to quell a time-sensitive life-threatening infection, and cause no harm in that the results from AST (whether furnished by SpecifAST or other method) of the subcultured samples will be in any event available as fast as the current methods allow.

Reviewer 2 Report

Dear authors, I really appreciate that the topic and presented research is of high importance and early detection of infectious agents are crucial in its management. The presented detection method using bacterial VOC is novel diagnostic approach and if proven more reliable and accurate than existing diagnostics methods, it will benefit clinicians with real time data at the bedside. The manuscript entitled "Susceptibility testing by volatile organic compound detection direct from positive blood cultures: a proof-of-principle laboratory study" is very well written and findings are presented clearly. 

However, few concerns from the manuscript are noted below and if provided those in details, it will help readers and other researchers to understand the significance of the study and approaches used therein.

  1. In the introduction, it is imperative to highlight the available reports on detection of VOCs in blood samples using CSA method. Just to highlight, there was a study published in 2017 using a CSA to detect microbial VOC from blood cultures (DOI: https://doi.org/10.1371/journal.pone.0173130). Kindly include a note on available reports in introduction. 
  2. Under Materials section, procedure for CSA using SpecifAST need to be presented in details. Presently, there are no such details are given in the manuscript.
  3. There is need to include few additional details such as minimal CFUs required to detect bacterial VOCs, specificity and sensitivity of this assay. 
  4. Under section 2.3, authors have mentioned for use of different antibiotics for each strain included. Is there any rational for selection of these antibiotics?
  5. Under section 2.4, on line 120, author states that VOC patterns were measured every 10 minutes. How many total measurements were recorded for each samples is missing. Please specify.
  6. On page number 8, line number 230, author have mentioned abouts factors that can influence the VOC levels. Thus, which measures were  taken in this study to avoid influence of such factors on readings?

Author Response

Dear authors, I really appreciate that the topic and presented research is of high importance and early detection of infectious agents are crucial in its management. The presented detection method using bacterial VOC is novel diagnostic approach and if proven more reliable and accurate than existing diagnostics methods, it will benefit clinicians with real time data at the bedside. The manuscript entitled "Susceptibility testing by volatile organic compound detection direct from positive blood cultures: a proof-of-principle laboratory study" is very well written and findings are presented clearly.

However, few concerns from the manuscript are noted below and if provided those in details, it will help readers and other researchers to understand the significance of the study and approaches used therein.

Reviewer #2 Question #1: In the introduction, it is imperative to highlight the available reports on detection of VOCs in blood samples using CSA method. Just to highlight, there was a study published in 2017 using a CSA to detect microbial VOC from blood cultures (DOI: https://doi.org/10.1371/journal.pone.0173130). Kindly include a note on available reports in introduction.

Reviewer #2 Answer #1: Thank you for this suggestion. We have added this study to our introduction and expanded the introduction section on this topic, lines 102 – 103:

Also, identification of different yeast species was described previously with promising results

Reviewer #2 Question #2: Under Materials section, procedure for CSA using SpecifAST need to be presented in details. Presently, there are no such details are given in the manuscript.

Reviewer #2 Answer #2: Details regarding the CSA caps and a basic explanation of the mechanism has been added, lines 157-163:

Once the vials were inoculated with the bacterial suspension, a pre-assembled CSA cap was screwed on top of the vial. The cap contained six indicator spots which react to VOCs via color change in the presence of bacterial growth. Samples were stationaryThe vials were inverted on a flat-bed scanner with the CSA caps facing the scanner. This assembly was  incubated at 37°C at 200 rpm, and VOC patterns were measured every 10 minutes. The total run time was 12 hours collecting a total of 72 images by the completion of the assay.”

Reviewer #2 Question #3: There is need to include few additional details such as minimal CFUs required to detect bacterial VOCs, specificity and sensitivity of this assay.

Reviewer #2 Answer #3: We did not determine the minimal CFU to detect bacterial VOCs as this was not within the scope of this study. The intended role for the SpecifAST was rapid AST after bacterial growth detection in blood culture bottles. As growth is detected there is a significant amount of CFU per blood culture bottle. The lowest CFU/mL we used was 5*106 CFU/mL (blood culture bottle) diluted to 5000 CFU in the SpecifAST test vial. As stated in our discussion section – although not formerly tested – we did not see any association between number of CFU and concordance of susceptibility results (lines 270 - 281).

We have included the sensitivity and specificity with corresponding 95%-CI to the results section and the detailed sensitivity and specificity per antibiotic is added to the supplementary file (Supplementary table 4), lines 225 – 228:

The overall sensitivity and specificity of the SpecifAST method compared to the VITEK2 was 91% (95%-CI: 83.1 - 98.9) and 97% (95%-CI: 94.8 – 99.2), Supplementary table 4.”

Reviewer #2 Question #4: Under section 2.3, authors have mentioned for use of different antibiotics for each strain included. Is there any rational for selection of these antibiotics?

Reviewer #2 Answer #4: The SpecifAST system was a preterm system with limited test capacity (in contrast to the further developed RevealAST system). Therefore, we were restricted to a few antibiotic agents. We decided to select antibiotics from different antibiotic classes, and antibiotics which are clinically relevant to treat patients with the corresponding micro-organisms in bacteremia. In our academic hospital those antibiotics are the preferred treatment options.

Reviewer #2 Question #5: Under section 2.4, on line 120, author states that VOC patterns were measured every 10 minutes. How many total measurements were recorded for each samples is missing. Please specify.

Reviewer #2 Answer #5: The assay ran for a total of 12 hours totaling 72 images being collected. This has been added to the manuscript. See lines 162 – 163:

The total run time was 12 hours collecting a total of 72 images by the completion of the assay.”

Reviewer #2 Question #6: On page number 8, line number 230, author have mentioned abouts factors that can influence the VOC levels. Thus, which measures were  taken in this study to avoid influence of such factors on readings?

Reviewer #2 Answer #6: These experiments were performed under controlled settings. CSA containing caps were packaged in Nitrogen with a desiccant and opened right before being used for the experiment. Any un-used CSA caps were discarded.  CSA response from the vials containing antibiotics was compared to the positive control (media and inoculum, no antibiotic) and negative control (media only, no inoculum), thus all CSA caps were exposed to same environmental conditions (temperature, humidity etc.) and same media volatiles.

Reviewer 3 Report

The paper is interesting and in my opinion it could be published in "Antibiotics" after making a minor revision  regarding the style of references.  In present version of the manuscript in some references the full titles of the Journals are used and in others the abbreviations of the titles of the Journals are presented - it should be corrected in order to obtain an uniform style.

Author Response

The paper is interesting and in my opinion it could be published in "Antibiotics" after making a minor revision  regarding the style of references.  In present version of the manuscript in some references the full titles of the Journals are used and in others the abbreviations of the titles of the Journals are presented - it should be corrected in order to obtain an uniform style.

Thank you for notifying. We have changed the endnote style to the preferred layout.